# Synthesis, Structural Characterization and Anticancer Activity of New 5-Trifluoromethyl-2-thioxo-thiazolo[4,5-*d*]pyrimidine Derivatives

**DOI:** 10.3390/ph15010092

**Published:** 2022-01-13

**Authors:** Lilianna Becan, Anna Pyra, Nina Rembiałkowska, Iwona Bryndal

**Affiliations:** 1Department of Organic Chemistry and Pharmaceutical Technology, Faculty of Pharmacy, Wroclaw Medical University, 211A Borowska, 50-556 Wrocław, Poland; 2Faculty of Chemistry, University of Wrocław, 14 Joliot-Curie, 50-383 Wrocław, Poland; anna.pyra@chem.uni.wroc.pl; 3Department of Molecular and Cellular Biology, Faculty of Pharmacy, Wroclaw Medical University, 211A Borowska, 50-556 Wrocław, Poland; nina.rembialkowska@umw.edu.pl

**Keywords:** trifluoromethyl derivatives, thiazolo[4,5-*d*]pyrimidines, anticancer activity

## Abstract

Thiazolo[4,5-*d*]pyrimidine derivatives are considered potential therapeutic agents, particularly in the development of anticancer drugs. In this study, new 7-oxo-(**2a-e**), 7-chloro-(**3a-e**) and also three 7-amino-(**4a-c**) 5-trifluoromethyl-2-thioxo-thiazolo[4,5-*d*]pyrimidine derivatives have been synthesized and evaluated for their potential anticancer activity. These derivatives were characterized by spectroscopic methods and elemental analysis, and the single-crystal X-ray diffraction was further performed to confirm a 3D structure for compounds **2e** and **4b**. The antiproliferative activity evaluation of twelve new compounds was carried out on a variety of cell lines including four human cancer (A375, C32, DU145, MCF-7/WT) and two normal cell lines (CHO-K1 and HaCaT). Four of them (**2b**, **3b**, **4b** and **4c**) were selected by the National Cancer Institute and evaluated for their in vitro anticancer activity using the NCI-60 screening program. 7-Chloro-3-phenyl-5-(trifluoromethyl)[1,3]thiazolo[4,5-*d*]pyrimidine-2(3*H*)-thione (**3b**) proved to be the most active among the newly synthesized compounds.

## 1. Introduction

Fluorinated organic molecules are of exceptional interest in medicinal chemistry [1,2,3]. The inclusion of fluorine in the molecule often results in improved pharmacological properties [4,5]. At present, fluorinated particles are an important class of anticancer drugs [6]. Fluorine, as a strongly electronegative small substituent, increases lipophilicity and also has an influence on steric effects that change the shape of the molecule. The fluorinated molecule also has greater stability because the C-F bond is stronger than the C-H bond [7,8]. Therefore, it is not quickly degraded by natural enzymes and the biotransformation, especially oxidation, of such substituted compounds is difficult [9,10]. With three fluorine atoms, trifluoromethyl moiety is very lipophilic functional group with an electronegative nature [11]. Fluorine is rarely found in natural organic compounds, but drugs containing the trifluoromethyl group are unexpectedly abundant. Drugs with a trifluoromethyl substituent are used in the treatment of many diseases, and as anticancer agents (Figure 1). In 2019, the FDA approved Alpelisib (A), a phosphatidylinositol 3-kinase (PI3K) inhibitor, for use in combination with fulvestrant (B) in the treatment of advanced or metastatic breast cancer [12]. Vilaprisan (C) [13], a synthetic and steroidal selective progesterone receptor modulator, is under investigation for the treatment of leiomyoma. Cevipabulin (D) [14], a polycyclic aromatic compound with potential antineoplastic activity, is a candidate for antitumor clinical trials. Flutamide (E) [15], a non-steroidal anti-androgen, is widely used for the treatment of prostate cancer and Sorafenib (F) [16], a multikinase inhibitor, has been shown to be effective in the cure for hepatocellular carcinoma. Here, we describe the structural modification of the 7-thia analogs of purines—thiazolo[4,5-*d*]pyrimidines—consisting of the introduction of a trifluoromethyl moiety into the 5-position of the scaffold. Thiazolo[4,5-*d*]pyrimidines, as a purine antagonist, exhibits potential biological activity, including anticancer activity [17,18,19,20,21]. It was expected that a combination of the trifluoromethyl group and thiazolo[4,5-*d*]pyrimidine structure in one molecule can create an active compound with good healing properties. According to the records, the introduction of a trifluoromethyl group in the structure could improve the bioavailability of new compounds. A novel series of 5-(trifluoromethyl)-2-thioxo-thiazolo[4,5-*d*]pyrimidines was synthesized and evaluated for its in vitro cytotoxicity against cancer cell lines. Twelve compounds (**2a-e**, **3a-d** and **4a-c**) were tested for their antiproliferative potential against four human cancer cell lines and two normal cell lines. In the study, a metabolic test determining mitochondrial activity was used, which determines the proliferative capacity of cells. Four of the prepared compounds, **2b**, **3b**, **4b** and **4c**, were selected by the National Cancer Institute (NCI) (Bethesda, MD, USA) under the Development Therapeutic Program for an in vitro screening program in the 60 human cancer cell lines.

## 2. Results and Discussion

### 2.1. Chemistry

#### 2.1.1. Synthesis and Physiochemical Properties

The preparation of 5-trifluoromethyl-thiazolo[4,5-*d*]pyrimidines is depicted in Figure 1.

The starting 4-amino-2-thioxo-2,3-dihydro-3-substituted-1,3-thiazole-5-carboxamides, **1a-e**, were prepared in a one-pot reaction from sulphur, 2-cyanoacetamide and appropriate isothiocyanate according to the procedure reported by Gewald [22]. The 3-substituted-2-thioxo-5-(trifluoromethyl)-2,3-dihydro[1,3]thiazolo[4,5-*d*]pyrimidin-7(*6H*)-ones **2a-e** were obtained using the previously described method [20,22] for the synthesis of 5-methyl derivatives. The source of a CF_3_ group in our synthesis was trifluoroacetic anhydride. The reaction of thiazole-5-carboxamides **1a-e** and the trifluoroacetic anhydride under heating resulted in pyrimidine ring formation and gave the desired thiazolo[4,5-*d*]pyrimidine core. In the IR spectra of the cyclized compounds, an absorption band in the region 1122–1166 cm^−1^ is observed due to stretching vibrations of the CF_3_ group. Characteristic bands of carbonyl occurred at 1661–1666 cm^−1^. ^1^H-NMR spectra of compounds **2a-c** exhibit the broad singlet signal attributed to the 6-NH proton, observed at δ 8.37, 12.98 and 13.85 ppm, respectively. In the ^1^H-NMR spectra of **2d** and **2e**, the NH proton signal is invisible. Next, chlorination of compounds **2a-e** with a mixture of phosphorus oxychloride and phosphorus pentachloride was performed at reflux and gave 7-chloro derivatives **3a-e**. The IR spectra lacked the carbonyl absorption peaks and, in the region 1700–1716 cm^−1^, an intense band is observed due to C=N stretching vibrations. The ^1^H-NMR spectrum for compound **3a** showed a doublet of quartets for the CH_2_ protons, due to complex coupling. The observed splitting resulted from the coupling of one set of hydrogens to just one set in their neighbor, perhaps due to the steric effects between the ethyl substituents and the fluorine and chlorine atoms in the structure of **3a**. Compounds **4a-c** were achieved by the nucleophilic substitution reaction of **3b** with an excess of the appropriate amine.

In all the 7-substituted-amino derivatives, **4a-c**, the infrared spectrum exhibited a signal confirming the presence of secondary amine group. The band was observed at 3277–3307 cm^−1^. The ^1^H-NMR spectra revealed the presence of a triplet signal at 8.92–8.56 ppm, attributed to the proton of this secondary amine. All newly synthesized compounds gave satisfactory spectral data from ^1^H- and ^13^C-NMR and IR spectroscopy, MS mass spectrometry, and elemental analyses (see Appendix A). The spectral data confirmed that the new compounds have the expected structures. 

The newly obtained compounds **2a-e**, **3a-e** and **4a-c** do not violate Lipinski’s “Rule of 5” for potential drug candidates [23]. The rules state that most molecules with a good bioavailability have a limited lipophilicity, expressed by logP ≤ 5 (octanol/water partition coefficient), a molecular weight MW ≤ 500, number of hydrogen bond donors ≤5, number of hydrogen bond acceptors ≤10. These parameters, calculated with Molinspiration, are depicted in Table 1 [24].

#### 2.1.2. Crystal Structures of the Compounds **2e** and **4b**

Each derivative of **2a-2e** can exist in two tautomeric forms (lactam–lactim) with an NH or OH group on the pyrimidine ring. Moreover, the ^1^H-NMR spectra of the **2d** and **2e** signal of the NH proton are invisible; therefore, additional confirmation was necessary. Despite our efforts, we were unable to obtain **3b** or any of the **three** monocrystalline derivatives that are suitable for studying the X-ray structure. Therefore, we took steps to obtain **2a-2e** derivatives and the last products (**4a-4c**) of the synthesis in crystalline form. This would enable an unambiguous confirmation of the assumed course of the reaction, and additionally determine which tautomeric form is preferred in the solid state. Compounds **2e** and **4b** were obtained as researchable single crystals; their structures were determined by X-ray crystallography and adopted as a structural model for compounds of groups **2** and **4**, respectively.

The X-ray diffraction data analysis indicated that 3-(4-fluoropheny)l-2-thioxo-5-(trifluoromethyl)-2,3-dihydro[1,3]thiazolo[4,5-*d*]pyrimidin-7(6H)-one (**2e**) crystallized as a semi-solvate in the triclinic space group *P*–1. The asymmetric unit was composed of one 2e molecule (Figure 2) in the lactim form and half of a toluene molecule, which was disordered about the center of symmetry. The thiazolopyrimidine fragment was nearly coplanar and the 4-fluorophenyl ring bound to N3 was rotated with respect to the mean plane of the central ring by 64.4 (1)°. In the crystal structure, the molecules of 2e were linked into a centrosymmetric dimer by N6-H6···O7^i^ hydrogen bond [H···A = 1.83 Å, D···A = 2.707 (2) Å, D−H···A = 173°, (i) −*x* + 1, −*y* + 2, −*z* + 2], involving the atom H6 of the pyrimidine NH group as a donor and the carbonyl O7 atom as an acceptor (Figure 3). Such dimers are further linked via rather weak C-H···S [C32-H32···S2^ii^, H···A = 2.82 Å, D···A = 3.732 (2) Å, D−H···A = 160°, (ii) −*x*, −*y* + 2, −*z* + 1] and C-H···F [C33—H33···F51^iii^, H···A = 2.58 Å, D···A = 3.480 (3) Å, D−H···A = 157°, (iii) −*x*, −*y* + 1, −*z* + 1] interactions (Appendix A). Analysis using PLATON program [25] shows that the aromatic π–π stacking interactions were also recognized in the **2e** crystal, as may be seen by the comparatively short distance between the centroids of the 4-fluorophenyl [C31-C36 (Cg1)] rings from adjacent molecules [Cg1∙∙∙Cg1^iii^ = 3.8861(17) Å].

Compound **4b** crystallized in the *P*2_1_/*n* space group with one molecule in the asymmetric unit (Figure 4). Both the thiazolopyrimidine and the phenyl rings were flat and formed a dihedral angle of 70.8 (1)° to each other. In the crystal of **4b**, N7–H7∙∙∙S2^i^ hydrogen bonds involving the amine N7 and the thioxo S2 atoms [H···A = 2.52 Å, D···A = 3.287 (3) Å, D−H···A = 147°, (i) −*x* + 1/2, *y* + 1/2, −*z* + 3/2] linked the molecules into zig-zag chains running along the ***b***-axis (Figure 5). The interchain contacts were provided by weak C—H···S [C36—H36···S2^iii^, H···A = 2.75 Å, D···A = 3.676 (4) Å, D−H···A = 165°, (iii) *x*, *y* + 1, *z*] and C—H···F bonds [C32—H32···F51^ii^, H···A = 2.62 Å, D···A = 3.306 (3) Å, D−H···A = 129°, (ii) *x*, *y* − 1, *z*]. In addition, the presence of C—H···π and π···π interactions completed the rise of the 3D network.

### 2.2. Biological Activity

#### 2.2.1. Antiproliferative Activity

Twelve compounds (**2a-e**, **3a-d** and **4a-c**) at a concentration range of 50–5000 μM were tested for their antiproliferative potential against four human cancer cell lines (A375—melanotic melanoma, C32—amelanotic melanoma, DU145—prostate cancer androgen-independent, MCF-7/WT—breast adenocarcinoma) and two normal cell lines: Chinese Hamster Ovary (CHO-K1) and immortalized human keratinocytes from histologically normal skin (HaCaT). The results obtained from the exposition of cancer and normal cells to newly synthesized compounds are shown in Figure 6.

The data exhibit the excellent antiproliferative activity of compounds **3a-3d** and **4a**. The strongest cytotoxic effect was observed after 72 h incubation of compound **3b** for each of the tested cancer cell lines and for normal cells (HaCaT). The lowest tested concentration (50 µM) of compound **3b** reduced the viability to 20 % for melanotic and amelanotic melanoma cell lines and human keratinocytes. A two times higher concentration was seen, as 100 µM of compounds from the **3a-3d** group also caused a significant decrease in viability. A strong cytotoxic effect was also observed for **3a** in C32, DU145 cells and human keratinocytes; for **3c** in C32, CHO-K1 cells and human keratinocytes; and for **3d** in C32 cells. The results showed that high cytotoxicity was obtained in DU145 and CHO-K1 cells after 72 h of incubation with **4a**. The most sensitive cell line to the action of tested compounds was DU145, and the most resistant line was MCF-7/WT.

The studied compounds were subjected to the determination of the median growth inhibitory concentration (IC_50_). Table 2 illustrates the IC_50_ values of compounds **2a-2e**, **3a-3d**, **4a-4c** toward A375, C32, DU145, MCF-7/WT, CHO-K1 and HaCaT cell lines. The data collected in Table 2 exhibit the excellent growth inhibition of compound **3b** against melanotic and amelanotic melanoma, breast, prostate cancer and keratinocytes cells. The strongest cytotoxic effect was observed for both types of melanoma cancer cells (C32; IC_50_ = 24.4 µM; A375; IC_50_ = 25.4 µM). In case of normal cells (CHO-K1 and HaCaT), IC_50_ = 75.5 µM and IC_50_ = 33.5 µM were acquired, respectively. The most active compounds, **3a-3d,** should be further tested to distinguish the type of cell death. One of the important criteria for a cancer therapeutic drug is that its side effects on normal cells in the body are minimized. Our results indicated that compounds **3a**, **3b** and **3c** caused a significant decrease in cell viability in both melanoma lines and normal keratinocytes. This effect has not been observed in the case of normal ovarian cells. Interestingly, keratinocytes and melanomas are skin cells and have revealed a higher sensitivity to experimental treatments. This suggests the selectivity of compounds **3a**, **3b** and **3c** for this type of cell. This may be because human skin cells are more susceptible than endodermal cells. Compound **3d**, active against both melanoma lines (C32; IC_50_ = 87.4 µM; A375; IC_50_ = 103 µM), proved to be non-toxic to normal cells (CHO-K1 and HaCaT). The largest difference in effect on normal cells was observed for compound **4a**: CHO-K1 and HaCaT, IC_50_ = 48.5 and 747.5 µM, respectively. Different cellular responses underlie differences in the action of carcinogenic compounds in different tissues.

The lowest values of the mean A (Table 2) refer to 3b, 3c and 3a, respectively. Of all the compounds tested, these three compounds showed the highest toxicity. The lowest value of the mean B occurred for the DU-145 cell line, which showed the highest sensitivity to compounds. The highest value was seen for the normal HACAT line, which may indicate that it is not as susceptible to the effects of the test compounds.

#### 2.2.2. Anticancer Screening Data Analysis

The search for the cytotoxic activity of new derivatives usually starts with in vitro cell-based screening. Four compounds **2b** (NSC D-745983/1), **3b** (NSC D-766111/1), **4b** (NSC D-785591/1) and **4c** (NSC D-771706/1), selected by the National Cancer Institute were tested in the anticancer screen for a panel of approximately 60 human disease-oriented tumor cell lines, organized into subpanels derived from nine human cancer types—leukemia, melanoma, and cancers of the lung, colon, brain, breast, ovary, prostate, and kidney [26]. They were initially tested at only one concentration of 10 µM (10^−5^ M) for each cell line, which were inoculated and preincubated on a microtiter plate for 48 h. The results of the one-dose anticancer assay for the tested new compounds are reported as the percentage of the treated cell growth in comparison with untreated no-drug control. This allows for detection of the growth inhibition—values 0–100 and cytotoxicity; negative numbers mean cell death. Derivatives which significantly reduced the growth in the full panel of cell lines were subsequently passed on for evaluation in the main test at a tenfold dilution of five different concentrations, from 10^−4^ to 10^−8^ M [27,28,29].

Table 3 represents a general overview of anticancer activity for the screened derivatives. Compounds **2b**, **4b** and **4c** showed insufficient activity in the one-dose assay; they were devoid of threshold activity and were dropped from the screening program. Compound **2b** demonstrated selective antitumor activity against the individual Ovarian C. cell line IGROV1 (growth% −5.14). Compounds **4b** and **4c**, with 7-amino substituents, are inactive. A remarkably low mean growth percent value of 29.51 was obtained only for compound **3b**. Compound **3b**, which possesses electronegative atom Cl, demonstrated much higher average activity against tumor cell lines in comparison with 7-oxo and 7-substituted-amino derivatives. The most sensitive cell lines for this compound are leukemia CCRF-CEM, HL-60(TB) and MOLT-4 (growth% −51.41, −41.20 and −27.71, respectively), NS cell lung C. HOP-92 and NCI-H522 (growth% −21.23 and −67.57, respectively), colon C. HCT-116, HCC-2998 and SW620 (growth% −27.21, −26.98 and −63.05, respectively), melanoma SK-MEL-28 (growth% −62.53), ovarian C. OVCAR-3 (growth% −41.27), renal C. UO-31 and RFX (growth% −82.97 and −16.27) and breast C. T-47D (growth% −35.57).

The importance of the 5-trifluoromethyl group on thiazolo[4,5-*d*]pyrimidine ring was explored by comparing the antiproliferative activity of the newly screened compounds **2b**, **3b**, **4b** and **4c**, with our previously obtained compounds, which were tested in the NCI, and have a similar structure, as the closest structural analogs **I**–**VII** (Figure 7) [30].

Table 4 presents an overview of antitumor activity in one-dose 10^−5^ M. 7-Oxo derivatives with trifluoromethyl **2b**, heterocyclic **III** and aromatic **I**, **II.** Substituents in the 5-position showed very low activity, with a mean growth percent of tumor cell lines equal to 80–106%, but 3-phenyl-2-thioxo-5-(trifluoromethyl)-2,3-dihydro[1,3]thiazolo[4,5-*d*]pyrimidin-7(6*H*)-one **2b** was more active than its analogs **I**-**III** against one single line, IGROV1 (growth% −5.14). The insertion of a chlorine atom in position 7 of the thiazolo[4,5-*d*]pyrimidine scaffold led to increased activity in these compounds compared to 7-oxo derivatives. The mean growth percent for compounds **3b**, **IV** and **V** decreased to a value of 20-64%. The derivative **IV** containing heterocyclic pyridine at the 5-position was the most active in this group. The most sensitive to these 7-chloro derivatives were renal cancer cell lines UO-31 for **3b** and **IV**, and CAKI-1 for **V** (growth% −82.97, −85,59 and −88.95, respectively). Among the compounds with 7-[(4-fluorobenzyl)amino)] moiety **4c**, **VI** and **VII,** only structural analog **VI** revealed significant activity against the full panel, with a mean growth percent of 48.46%. It could be that the presence of chlorine atom in the 4-chlorophenyl moiety improves activity.

Antitumor activity in five dose testing (10^−4^−10^−8^ M) is expressed by three different dose–response parameters for each of the 60 human tumor cell lines. The GI_50_ value (molar concentration causing a half-growth inhibition), TGI (molar concentration required for total growth inhibition), which signifies the cytostatic effect, and LC_50_ (molar concentration of the compound required for 50% cell death). LC_50_ signifies the cytotoxic effect. A mean graph midpoint (MG_MID) was calculated for each of these parameters and displays an averaged activity parameter over the full panel of screened tumor cell lines. Table 3 shows the antiproliferative data for four tested compounds **2b**, **3b**, **4b** and **4c** against 60 human cancer cell lines in one dose and main parameters that characterize cytostatic activity for the most active compound **3b**. This compound was passed on for extensive evaluation over a five-log dose range and showed very good antitumor properties against all the tested subpanel tumor cell lines at the GI_50_ and TGI levels, and most at the LC_50_. 7-Chloro derivative **3b** showed moderate selectivity on all 60 cell lines; the values of Δ log for three parameters are only slightly larger than 1. Nevertheless, the selectivity for GI_50_ can be seen for several cell lines within subpanel leukemia (SR, log GI_50_ −6.36) and non-small cell lung cancer (NCI-H522, log GI_50_ −6.69). Compound **3b** demonstrated the highest cytotoxic activity for the melanoma, CNS and renal cancer among the tested subpanels tumor cell lines at the LC_50_ indicator. Table 5 demonstrates an overview of the main parameters that characterize cytostatic activity for the most active new compound, **3b**, and its analogs, **IV** and **V**.

As can be seen from Table 1, the 7-chloro active compounds **3a-3e** have greater lipophilicity than their 7-oxo analogs **2a-2e** However, the inactive compound **4c** is also characterized by high lipophilicity, i.e., it cannot be a property that determines the activity for this group of compounds. From the analysis of the data reported in Table 3, Table 4 and Table 5, we can evince that chlorine atom on the 7 position of thiazolo[4,5-*d*]pyrimidine ring, rather than a substituent on the 5 position, appears to favorably modulate anticancer activity. The change in the chlorine atom from the 7-position to an amino group leads to the inactive compounds **4b** and **4c**. The comparison of the main parameters that characterize the cytostatic activity of the tested compounds, trifluoromethyl **3b** and its analogs **IV** and **V** from our previous paper, revealed a number of observations. As shown in Table 4, the mean percent growth of tumor cell lines at a dose of 10^−5^ M for analog **IV** and the trifluoromethyl derivative **3b** is comparable. Nevertheless, the trifluoromethyl derivative **3b** exhibited the highest average anticancer activity (MIG_MID log GI_50_ −5.66, logTGI −5.30, log LC_50_ −4.38) and demonstrated cytostatic activity against all 58 tested cell lines (GI_50_, TGI) and cytotoxic effects relative to 45 cell lines (LC_50_) tested with it. Analog **V** showed good results, with 58, 49 and 35 cell lines, whereas **IV** demonstrated cytotoxicity (LC_50_) against only 3 of the 59 tested cell lines. From these data, collected in Table 5, the cytotoxic activity of the trifluoromethyl derivative **3b** is shown to be comparable to its structural analogs **IV** and **V**. The trifluoromethyl substituent does not have as much influence on the activity of the thiazolo[4,5-*d*]pyrimidine derivatives as was assumed.

## 3. Materials and Methods

### 3.1. Chemistry

Melting points were determined on a Mel-Temp apparatus and are uncorrected. The IR spectra were recorded on a Nicolett FT-IR spectrometer and are reported in cm^−1^. The ^1^H- and ^13^ C-NMR spectra were recorded on a BRUKER ARX 300 MHz instrument using dimethyl sulfoxide-d6 or CDCl_3_ as a solvent. All chemical shifts are reported in a ppm down field relative to the chemical shifts in teramethylsilane. The progress of the reaction and the purity of obtained compounds were monitored by thin-layer chromatography (TLC) on Merck silica gel plates (Merck F_254_, Darmstadt, Germany), using the solvent system ethyl acetate:hexane 1:1 for eluation. Elemental analyses were performed on a Perkin Elmer analyzer (Waltham, MA, USA); the results are within ±0.4% of the calculated values. Molecular weight of final compounds was assessed by electrospray ionization mass spectrometry from Bruker Daltonics (Billerica, MA, USA). The reagents and chemicals for synthesis were purchased from commercial sources and used with no further purification from Alfa Aesar (Ward Hill, MA, USA), Sigma Aldrich (Burlington, MA, USA) or Chempur (Bangalore, India).

#### 3.1.1. Preparation of Compounds **1a-1e**

The starting thiazoles **1a-e** were prepared according to known procedures, described in references [20,22,30]. Appropriate isothiocyanate (100 nM) was added dropwise to a stirred suspension of cyanoacetamide (8.4 g, 100 mM), sulphur (3.2 g, 100 mM) and trimethylamine (12 mL) in ethanol (100 mL). Next, the mixture was stirred at 50–60 °C for 1 h and, after cooling, the precipitated product was filtered, washed with cold ethanol and recrystallized from glacial acetic acid.

*4-amino-3-ethyl-2-thioxo-2,3-dihydro-1,3-thiazole-5-caboxamide* (**1a**). Grey needles; Yield 87%, m.p. 182–183 °C [30]. 

*4-amino-3-phenyl-2-thioxo-2,3-dihydro-1,3-thiazole-5-carboxamide* (**1b**). Pale yellow needles; Yield 67%, m.p. 247–248 °C [22]. 

*4-amino-3-(2-fluorophenyl)-2-thioxo-2,3-dihydro-1,3-thiazole-5-caboxamide* (**1c**). Yellow needles; Yield 67%, m.p. 237–238 °C. ^1^H-NMR (DMSO) δ: 7.65–7.40 (m, 4H), 7.11 (s, 2H, NH_2_), 6.95 (s, 2H, NH_2_). 

*4-amino-3-(3-fluorophenyl)-2-thioxo-2,3-dihydro-1,3-thiazole-5-caboxamide* (**1d**). Yellow needles; Yield 67%, m.p. 222–223 °C. ^1^H-NMR (DMSO) δ: 7.66–7.20 (m, 4H), 7.07 (s, 2H, NH_2_), 6.85 (s, 2H, NH_2_). 

*4-amino-3-(4-fluorophenyl)-2-thioxo-2,3-dihydro-1,3-thiazole-5-caboxamide* (**1e**). Yellow needles; Yield 84%, m.p. 270–272 °C [20].

#### 3.1.2. Preparation of Compounds **2a-2e**

For the cyclocondensation reaction, a mixture of appropriate thiazole **1a-e** (10 mM) and 20 mL of trifluoroacetic anhydride was heated at reflux with stirring for 4 h and was kept at room temperature overnight. The precipitate was separated by filtration and then recrystallized from toluene.

*3-ethyl-2-thioxo-5-(trifluoromethyl)-2,3-dihydro*[1,3]*thiazolo[4,5-*d*]pyrimidin-7(6H)-one*

(**2a**). Orange solid, yield 62%, m.p. 229–230 °C. ^1^H-NMR (DMSO) δ: 8.37 (s, 1H, NH), 4.34 (q, *J* = 21.3 Hz, 2H, CH_2_), 1.26 (t, *J* = 14.1 Hz, 3H, CH_3_); ^13^C-NMR (DMSO) δ: 12.26 (x2C), 107.06, 117.64, 121.29, 159.02, 161.83, 189.21. IR cm^−1^: 3364, 1661, 1548, 1122, 739. MS (ESI, *m*/*z*) [M-H]^–^ calcd. for C_8_H_6_F_3_N_3_OS_2_ 279.9821; found 279.9891. Anal. calcd. C 34.16, H 2.15, N 14.94; found C 34.09, H 2.30, N 14.68%.

*3-phenyl-2-thioxo-5-(trifluoromethyl)-2,3-dihydro*[1,3]*thiazolo[4,5-*d*]pyrimidin-7(6H)-one* (**2b**). Yellow solid, yield 74%, m.p. 262–263 °C. ^1^H-NMR (DMSO) δ: 12.98 (s, 1H, NH), 7.61–7.37 (m, 5H, Ar-H); ^13^C-NMR (DMSO) δ: 106.73, 117.03, 120.68, 128.60, 129.51, 129.79, 135.14, 151.61, 152.10, 159.76, 161.35, 190.04. IR cm^–1^: 3050, 1662, 1587, 1146, 732. MS (ESI, *m*/*z*) [M-H]^–^ calcd. for C_12_H_6_F_3_N_3_OS_2_ 327.9821; found 327.9853. Anal. calcd. C 43.77, H 1.84, N 12.76; found C 44.09, H 1.75, N 12.48%.

*3-(2-fluoropheny)l-2-thioxo-5-(trifluoromethyl)-2,3-dihydro*[1,3]*thiazolo[4,5-*d*]pyrimidin-7(6H)-one* (**2c**).Yellow solid, yield 66%, m.p. 211–212 °C. ^1^H-NMR (DMSO) δ: 13.08 (s, 1H, NH), 7.69–7.40 (m, 4H, Ar-H); ^13^C-NMR (DMSO) δ: 107.52, 117.45, 121.11, 122.94, 126.04, 131.44, 133.08, 151.42, 152.45, 155.79, 162.13, 190.44. IR cm^–1^: 3433, 3321, 1654, 1596, 1154, 733. MS (ESI, *m*/*z*) [M-H]^–^ calcd. for C_12_H_5_F_4_N_3_OS_2_ 345.9737; found: 345.9887. Anal. calcd. C 41.50, H 1.45, N 12.10; found C 41.88, H 1.75, N 12.47%.

*3-(3-fluorophenyl)-2-thioxo-5-(trifluoromethyl)-2,3-dihydro*[1,3]*thiazolo[4,5-*d*]pyrimidin-7(6H)-one* (**2d**). Yellow solid, yield 61%, m.p. 263–264 °C. ^1^H-NMR (DMSO) δ: 7.70–7.12 (m, 4H, Ar-H); ^13^C-NMR (DMSO) δ: 107.17, 116.93, 125.76, 128.65, 129.35, 131.74, 136.92, 160.44, 160.91, 161.98, 164.15, 190.82. IR cm^–1^: 3052, 1666, 1599, 1153, 729. MS (ESI, *m*/*z*) [M-H]^–^ calcd. for C_12_H_5_F_4_N_3_OS_2_ 345.9726; found 345.9832. Anal. calcd. C 41.50, H 1.45, N 12.10; found C 41.79, H 1.68, N 12.32%.

*3-(4-fluoropheny)l-2-thioxo-5-(trifluoromethyl)-2,3-dihydro*[1,3]*thiazolo[4,5-*d*]pyrimidin-7(6H)-one* (**2e**). Yellow solid, yield 63%, m.p. 259–260 °C. ^1^H-NMR (CDCl_3_) δ: 7.32–7.17 (m, 4H, Ar-H); ^13^C-NMR (CDCl_3_) δ: 116.83, 117.15, 121.03, 125.30, 128,23, 129.04, 130.23, 130.35, 133.31, 137.88, 142.32, 160.17. IR cm^–1^: 3055, 1668, 1556, 1166, 732. MS (ESI, *m*/*z*) [M-H]^–^ calcd. for C_12_H_5_F_4_N_3_OS_2_ 345.9726; found 345.9803. Anal. calcd. C 41.50, H 1.45, N 12.10; found C 41.16, H 1.62, N 12.47%.

#### 3.1.3. Preparation of Compounds **3a-3e**

For the chlorination reaction, the appropriate 3-substituted-2-thioxo-5-(trifluoromethyl)-2,3-dihydro[1,3]thiazolo[4,5-*d*]pyrimidin-7(6*H*)-one **2a-e** (10 mM) was added to a solution of phosphorus pentachloride (2.08 g, 10 nM) in 20 mL of phosphorus oxychloride. The mixture was heated under reflux for 2 h and, after cooling, poured into 200 mL of ice-water. Then, it was filtered, washed with water, dried, and recrystallized from glacial acetic acid.

*7-chloro-3-ethyl-5-(trifluoromethyl)*[1,3]*thiazolo[4,5-*d*]pyrimidine-2(3H)-thione* (**3a**). Yellow solid, yield 60%, m.p. 66–67 °C. ^1^H-NMR (DMSO) δ: 4.42–3.97 (dq, *J* = 21.6, *J* = 21.6, 2H, CH_2_), 1.27 (t, *J* = 19.8 Mz, 3H, CH_3_); ^13^C-NMR (DMSO) δ: 11.92, 12.86, 107.04, 117.53, 121.17, 150.13, 158.02, 166,80, 189.35. IR cm^–1^: 1709, 1548, 1343, 1154, 739. MS (ESI, *m*/*z*) [M-H]^–^ calcd. for C_8_H_5_ClF_3_N_3_S_2_ 298.9482; found 298.9494. Anal. calcd. C 32.06, H 1.68, N 14.02; found C 32.10, H 1.84, N 14.25%.

*7-chloro-3-phenyl-5-(trifluoromethyl)*[1,3]*thiazolo[4,5-*d*]pyrimidine-2(3H)-thione* (**3b**). Yellow solid, yield 72%, m.p. 132–133 °C. ^1^H-NMR (DMSO) δ: 7.67–7.50 (m, 5H, Ar-H); ^13^C-NMR (DMSO) δ: 127.74, 128.34, 129.54, 129.79, 129.89, 130.26, 132.58 134.52, 150.48, 158.11, 166.34, 190.23. IR cm^–1^: 1700, 1541, 1342, 1150,732. MS (ESI, *m*/*z*) [M-H]^–^ calcd. for C_12_H_5_ClF_3_N_3_S_2_ 345.9482; found 345.9508. Anal. calcd. C 41.44, H 1.45, N 12.08; found C 41.56, H 1.54, N 11.93%.

*7-chloro-(2-fluoropheny)-5-(trifluoromethyl)*[1,3]*thiazolo[4,5-*d*]pyrimidine-2(3H)-thione (**3c**)*. Yellow solid, yield 56%, m.p. 140–141 °C. ^1^H-NMR (DMSO) δ: 7.65–7.43 (m, 4H, arom.); ^13^C-NMR (DMSO) δ: 117.44, 117.67, 126.08, 126.32, 128.65, 129.35, 131.46, 151.42, 159.58, 162.23, 164.01, 190.43. IR cm^–1^: 1712, 1542, 1115, 734. MS (ESI, *m*/*z*) [M-H]^–^ calcd. for C_12_H_4_ClF_4_N_3_S_2_ 363.9388; found 363.9448. Anal. calcd. C 39.41, H 1.10, N 11.49; found C 39.64, H 1.20, N 11.88%.

*7-chloro-(3-fluoropheny)-5-(trifluoromethyl)*[1,3]*thiazolo[4,5-*d*]pyrimidine-2(3H)-thione* (**3d**). Yellow solid, yield 72%, m.p. 128–129 °C. ^1^H-NMR (DMSO) δ: 7.73–7.36 (m, 4H, Ar-H); ^13^C-NMR (DMSO) δ: 116.60 117.90, 124.71, 132.14, 134.12, 136.05, 151.73, 158.35, 160.73, 163.99, 166.57, 190.54. IR cm^–1^: 1716, 1544, 1194, 730. MS (ESI, *m*/*z*) [M-H]^–^ calcd. for C_12_H_4_ClF_4_N_3_S_2_ 363.9388; found: 363.9478. Anal. Calc: C,39.41; H,1.10; N,11.49%. Found C,39,47; H,0.98; N,11.91%.

*7-chloro-(4-fluoropheny)-5-(trifluoromethyl)*[1,3]*thiazolo[4,5-*d*]pyrimidine-2(3H)-thione* (**3e**). Yellow solid, yield 72%, m.p. 147–148 °C. ^1^H-NMR (DMSO) δ: 7.63–7.44 (m, 4H, Ar-H); ^13^C-NMR (DMSO) δ: 117.38, 117.52, 120.08, 129.23, 130.61, 150.98, 151.71, 158.62, 161.19, 164.46, 166.84, 190.89. IR cm^–1^: 1711, 1542, 1115, 734. MS (ESI, *m*/*z*) [M+H]^+^ calcd. for C_12_H_4_ClF_4_N_3_S_2_ 363.9388; found 363.9498. Anal. calcd. C 39.41, H 1.10, N 11.49; found C 39.34, H 0.95, N 11.43%.

#### 3.1.4. Preparation of Compounds **4a-4c**

For the reaction of **3b** with amines, the selected amine (20 mM) was added to a suspension of **3b** (3.45 g, 10 mM) in 20 mL of buthan-1-ol. The mixture was refluxed for 3 h, then cooled, and the obtained solid was filtered and recrystallized from buthan-1-ol.

*7-(methylamino)-3-phenyl-5-(trifluoromethyl)*[1,3]*thiazolo[4,5-*d*]pyrimidine-2(3H)-thione* (**4a**). Yellow solid, yield 65%, m.p. 197–198 °C. ^1^H-NMR (DMSO) δ: 8.56 (t, 1H, NH), 7.58–7.40 (m, 5H, Ar-H), 2.98 (d, *J* = 4.5 Hz, 3H, CH_3_); ^13^C-NMR (DMSO) δ: 28.05, 103.02, 117.38, 128.50, 129.17, 129.65, 129.86, 130.05, 135.89, 155.79, 168.36, 175.09, 190.21. IR cm^–1^: 3257, 3067, 1593, 1246, 1140. MS (ESI, *m*/*z*) [M-H]^–^ calcd. for C_13_H_9_F_3_N_4_S_2_ 341.0137; found 341.0232. Anal. calcd. C 45.61, H 2.65, N 16.36; found C 45.76, H 2.54, N 16.55%.

*7-(ethylamino)-3-phenyl-5-(trifluoromethyl)*[1,3]*thiazolo[4,5-*d*]pyrimidine-2(3H)-thione* (**4b**). Yellow solid, yield 70%, m.p. 217–218 °C. ^1^H-NMR (DMSO) δ: 8.81 (t, 1H, NH), 7.59–7.39 (m, 5H, Ar-H), 3.06–3.47 (q, 2H, CH_2_) 1.21-1.16 (t, 3H, CH_3_); ^13^C-NMR (DMSO) δ: 14.64, 36.19, 129.17, 129.54, 129.63, 129.84, 130.01, 133.85, 135.93, 153.23, 153.71, 155.08, 155.80, 190.37. IR cm^–1^: 3277, 2977, 2947, 2605, 1245, 1144, 734. MS (ESI, *m*/*z*) [M-H]^–^ calcd. for C_14_H_11_F_3_N_4_S_2_ 355.0293; found 355.0358. Anal. calcd. C 47.18, H 3.11, N 15.72; found C 47.08, H 3.41, N 15.12%.

*7-[(4-fluorobenzyl)amino)]-3-phenyl-5-(trifluoromethyl)*[1,3]*thiazolo[4,5-*d*]pyrimidine-2(3H)-thione* (**4c**). Gray solid, yield 70%, m.p. 269–270 °C. ^1^H-NMR (DMSO) δ: 8.91 (t, 1H, NH), 7.57–7.12 (m, 9H, Ar-H),), 4.64 (d, *J* = 45.7 Hz, 2H, CH_2_); ^13^C-NMR (DMSO) δ: 43.72, 115.41, 115.69, 115.98, 129.58, 129.65, 130.21, 130.32, 130.80, 130.84, 131.74, 131.85, 133.84, 135.31, 135.35, 155.73, 160.223, 163.44, 168.25. IR cm^–1^: 3307, 2963, 1701, 1155, 732. MS (ESI, *m*/*z*) [M-H]^–^ calcd. for C_19_H_12_F_4_N_4_S_2_ 435.0356; found 435.0546. Anal. calcd. C 52.29, H 2.77, N 12.84; found C 47.08, H 3.41, N 15.12%.

#### 3.1.5. X-ray Structural Studies

Crystals of compound **2e,** suitable for single-crystal X-ray diffraction analysis, were grown by the slow evaporation of toluene at ambient temperature and in the presence of air.

Instead, to obtain **4b** crystals, the solid was dissolved in methanol and propan-1-ol was added at any time until the solution turned cloudy and was allowed to slowly evaporate for several days. Diffraction data were collected with Cu-K_α_ radiation (λ = 1.5418 Å) radiations (ω-scan modes) using a Rigaku Oxford Diffraction XtaLABSynergy-R DW diffractometer equipped with a HyPix ARC 150° Hybrid Photon Counting (HPC) detector. The data were measured at 100(2) K by using an Oxford Cryosystems open-flow nitrogen cryostat. Data collection, cell refinement, data reduction and analysis were carried out with the CrysAlisPro software package [31]. Multi-scan absorption correction was applied to the data. The structures were solved by direct methods, using SHELXS-97 [32], and refined by a full-matrix least squares technique on F2 with SHELXL-2013 (and also with SHELXL-2018) [33] with anisotropic thermal parameters for all non-H atoms. During the refinement for **2e**, toluene molecule was found to exhibit disorder about the center of symmetry. In both compounds, all H atoms were found in different Fourier maps and isotropically refine, but in the final refinement cycles, they were repositioned in their calculated positions and refined using a riding model in geometrically optimized positions, with C–H = 0.95 Å (in **4b** also 0.99 and 0.98 Å for CH_2_ and CH_3_), and U_iso_(H) = 1.2U_eq_© for CH (in **4b** also 1.2©(C) for CH_2_ and 1.5U_eq_(C) for CH_3_); with N–H = 0.88 Å, and U_iso_(H) = 1.2U_eq_(N), respectively. All figures were made using the DIAMOND program [34]. An analysis of the intra- and intermolecular interactions was performed using the program PLATON [25]. Selected X-ray single-crystal data and structure refinement details are summarized in Appendix A.

### 3.2. Biological Activity

#### 3.2.1. Antiproliferative Activity

##### Cell Lines

To determine the antiproliferative activity of newly synthesized compounds, the following cell lines were used: human melanotic melanoma cell line (A375), human amelanotic melanoma cell line (C32), human breast adenocarcinoma cell line (MCF-7/WT), human prostate cancer androgen-independent (DU145) and two normal cell lines: hamster ovarian (CHO-K1) and immortalized human keratinocytes from histologically normal skin (HaCaT). The cells were obtained from the cell line bank of the Department of Molecular and Cellular Biology, Wroclaw Medical University. A375, C32, MCF-7/WT, DU145 and HaCaT cells were grown in a monolayer cultured in Dulbecco’s Modified Eagle’s Medium (DMEM, Sigma-Aldrich, Poznań, Poland), supplemented with 10% of fetal bovine serum (FBS, Gibco, Paisley, UK) and 1% of antibiotics (penicillin/streptomycin, Sigma-Aldrich) and CHO-K1 were grown in HAM’s F-12 medium (Sigma-Aldrich), supplemented with 10% fetal bovine serum (FBS, Gibco) and 1% of antibiotics (penicillin/streptomycin, Sigma-Aldrich). Cell lines were cultured in the polystyrene flasks 25 or 75 cm^2^ (Thermo Fisher Scientific, Waltham, MA, USA), which were stored in 37 °C and 5% CO_2_ in a humidified 95% atmosphere (SteriCult, Thermo Scientific, Alab). The cell medium was exchanged twice a week. The exponentially growing cells were used throughout the experiments. The cells were rinsed with phosphate-buffered saline (PBS, Sigma-Aldrich) and trypsinized by Trypsin-EDTA 0.25% solution (Sigma-Aldrich).

##### Cell Toxicity Test

The influence of the compounds was performed in a monolayer culture on cancer and normal cells. Compounds were initially dissolved in DMSO; further dilutions of the compound were performed in Dulbecco’s Modified Eagle’s Medium supplemented with 10% FBS. Compounds were tested in a 50–5000 µM concentration range. In the in vitro experiments, the concentrations of DMSO in the final solution of the compounds were used 0.05%; 0.1%; 0.5%; 1%, respectively.

##### Cell Viability Assays

Cell viability was determined using the MTT colorimetric assay. All cell lines were seeded into 96-well plates to a density of 3 × 10^4^ cells/well. After 24 h, the culture supernatants were removed and appropriate dilutions of compounds in the culture medium (200 µL/well) were added to the cells’ monolayer cultures and incubated for an additional 72 h. After 72 h of exposition to the compounds and the removal of the culture medium, cell viability was determined by reducing the yellow dye 3-(4,5-*d*imethyl-2-thiazol)-2,5-diphenyl-2H-tetrazolium bromide (MTT assay, Sigma Saint Louis, MO, USA; In Vitro Toxicology Assay) to a blue formazan product. The absorbance of the resulting solutions in three duplicate experiments was measured at the wavelength of 560 nm in a microplate reader (EnSpire, Perkin Elmer, Kraków, Poland). MTT assay is a colorimetric method that estimates the rate of metabolism in viable cells. Absorbance is directly proportional to the number of viable cells. All the experiments were performed three times and the mean absorbance values were calculated. The methodology for the evaluation of the growth of human cancer and normal cells was described previously [35]. The cell viability in each group was expressed as a percentage of control (untreated) cells.

#### 3.2.2. Anticancer Screening Methodology

##### Primary Anticancer Assay and Determination of GI_50_, TGI and LC_50_ Values

The antitumor screening (NCI-60) was performed at the National Cancer Institute, Bethesda, USA. A total panel of approximately 60 human tumor cell lines derived from nine different cancer types (leukemia, non-small cell lung, colon, brain, melanoma, ovarian, renal, prostate and breast) formed the basis of this in vitro test. The origins and processing of the cell lines used in the study were described previously [26]. First, the compounds were evaluated in a primary anticancer assay at 10^−5^ M. Results for each tested compound are reported as the percent of growth of the treated cells. Compounds with significant growth inhibition of the cell lines were passed on for evaluation at five concentration levels (10^−4^, 10^−5^, 10^−6^, 10^−7^ and 10^−8^ M). The human tumor cell lines of the cancer screening panel were grown in an RMPI 1640 medium containing 5% fetal bovine serum and 2 mM of L-glutamine. Each cell line was inoculated and preincubated at 37 °C, 5% CO_2_, 95% air and 100% relative humidity for 24 h, before the addition of the test compound into 96-well microtiter plates in 100 µL at plating densities ranging from 5000 to 40,000 cells/well. The density of inoculum depends on the type of tumor cell and its growth characteristics. The screened compounds were solubilized in dimethyl sulfoxide at 400-fold the desired final concentration and stored frozen. At the time of the test agents’ addition, an aliquot of frozen concentrate was thawed and diluted to twice the desired final concentration with complete medium containing 50 µg/mL gentamicin. Next, the test compounds were evaluated at the four additional 10-fold dilutions to reach the final concentrations plus control. Aliquots of 100 µL of the different dilutions (100, 10, 1, 01, 001 µM/mL) of the tested compounds were added to microtiter wells, and the culture was incubated for 48 h under the same conditions. Cell were fixed in situ by the addition of 50 µL of cold 50% trichloroacetic acid and incubated for 60 min at 4 °C. The supernatant was discarded; plates were washed five times with water and air-dried. Sulforhodamine B at 0.4% in 1% acetic acid, 100 µL, was added to each well, then plates were incubated for 10 min at room temperature. Unbound dye was removed by washing five times with 1% acetic acid and air dried. The protein-bound stain was solubilized with a 10 µM trizma base and the absorbance was read on an automated plate reader. A wavelength of 515 nm was used to read optical densities. The cytotoxic effects were evaluated and results and dose–response parameters were calculated [27,28,29].

## 4. Conclusions

A series of the new 5-trifluoromethyl-2-thioxo-thiazolo[4,5-*d*]pyrimidines **2a-e** was synthesized by the cyclization of the 4-amino-2-thioxo-2,3-dihydro-3-substituted-1,3-thiazole-5-carboxamides with trifluoroacetic anhydride. In the next step chlorination of **3a-e** gave the 7-chloro derivatives **3a-e**. Additionally, three 7-amino derivatives **4a-c** were obtained by the treatment of compound **3b** with methyl-, ethyl- and fluorobenzylamine. The structures were determined by IR, ^1^H-NMR, ^13^C-NMR, MS and elemental analysis. Furthermore, compounds **2e** and **4b** were also studied by X-ray crystallography. Both compounds crystallized with one molecule in the asymmetric unit, but in two different space groups; thus, the packing of both crystals was varied. **2e** molecules form dimers via N-H···O hydrogen bonds while **4b** molecules form a zig-zag chain due to the presence of N-H···S hydrogen bonds. The crystal structures of both compounds are stabilized by weak C-H···F, C-H···S and π···π interactions.

The biological investigations (twelve compounds **2a-e**, **3a-d** and **4a-c**) assessed the antiproliferative properties against human cancer (A375, C32, DU145, MCF-7/WT) and normal (CHO-K1 and HaCaT) cell lines. An excellent antiproliferative activity was exhibited by compounds **3a-3d** and **4a**. The results revealed a decrease in cells treated with **3b**, indicating that proliferation inhibited cancer cells. In normal human keratinocytes and hamster ovarian, a smaller or larger decrease in proliferation was observed depending on the compound. It was also shown that compounds **3a-3d** (7-chloro) and **4a**, with a small amino substituent, had a higher activity against human prostate, melanoma and were significantly less sensitive in normal fibroblasts. The increased activity of compounds **3a-3d** in comparison to compounds **2a-2e** may result from their greater lipophilicity (Table 1) and the presence of a strongly electronegative substituent at position 7.

Four compounds, **2b**, **3b**, **4b** and **4c,** selected by the National Cancer Institute, were used in a panel of approximately 60 human disease-oriented tumor cell lines in the anticancer screening. Among the screened compounds, only **3b** demonstrated both cytostatic and cytotoxic activities, especially for leukemia and colon cancer cell lines.

New derivatives with a trifluoromethyl moiety were designed, with the hope of obtaining active compounds. Indeed, in the conducted research, the new derivatives **3a-3d** and **4a** were very active, especially compound **3b**. However, the trifluoromethyl moiety introduced into the structure of the new thiazolo [4,5-*d*] pyrimidine derivatives did not result in a significant increase in activity as compared to the derivatives with heterocyclic or aromatic substituents. The activity of the new compound **3b** is comparable to its structural analogs. Together, these data indicate that newly developed 5-trifluoromethyl-2-thioxo-thiazolo[4,5-*d*]pyrimidine derivatives may be promising agents for further research and development as a prospective inhibitors in human cancer cells.

## Data Availability

Data is contained within the article or Appendix A.

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
