# Peer review of "Synthesis, Structural Characterization and Anticancer Activity of New 5-Trifluoromethyl-2-thioxo-thiazolo[4,5-d]pyrimidine Derivatives"

_pharmaceuticals, 2022, doi:10.3390/ph15010092_

Round 1
Reviewer 1 Report
The authors have improved the manuscript. However, some critical discussions are still missing, and the major revision suggested by the reviewer has not been addressed correctly. Although this field of research is of great interest to the journal, the manuscript lacks in-depth data analysis and interpretation. Hence, it is not recommended for publication.
- The authors responded in detail justifying their hypothesis towards the need to determine the x-ray crystal structure of the small molecules. However, this discussion is lacking in section 2.1.2.
- SD values in Table 2 are missing.
- Terms like slightly less toxic do not indicate anything to the reader. This term is used multiple times in the manuscript.
- No statistical analysis was carried out to indicate that if the improvements in the biological activities are significant.
For eg., the authors mention that “Our results indicated that compounds 3b and 3c caused a significant decrease in cell viability in both melanoma lines and normal keratinocytes.” In fact, from table 2 compound 3a shows similar IC50 values.
As previously pointed out by the reviewer, the values compared by the authors (Table 4- 29.51% for 3b and 20.78 for IV) are comparable and not lower as the authors inferred.
Again, based on the results in Table 5, most values for compounds 3b, IV and V are also comparable. Unfortunately, the reviewer is not convinced that compound 3b shows significantly higher activity than its other analogs IV and V.
- Trifluoroacetic acid anhydride should be renamed as trifluoroacetic anhydride- line 86, 87.
Author Response
Dear Reviewer,
Thank you once again for the reviewing our contribution and giving us another possibility to improve our manuscript in the best possible way. All new the changes made were marked in purple in the main manuscript.
Responses to the Reviewer's comments
The authors have improved the manuscript. However, some critical discussions are still missing, and the major revision suggested by the reviewer has not been addressed correctly. Although this field of research is of great interest to the journal, the manuscript lacks in-depth data analysis and interpretation. Hence, it is not recommended for publication.
- This remark was taken into consideration and the manuscript was corrected according to the Reviewer's comments.
- The authors responded in detail justifying their hypothesis towards the need to determine the x-ray crystal structure of the small molecules. However, this discussion is lacking in section 2.1.2.
- This was corrected.
- SD values in Table 2 are missing.
-The values (SD) were added.
- Terms like slightly less toxicdo not indicate anything to the reader. This term is used multiple times in the manuscript.
- This sentence was removed. However, term “less toxic” was used only here.
- No statistical analysis was carried out to indicate that if the improvements in the biological activities are significant.
- The values of the mean IC50 for the given compound (Mean A) and for the cell lines (Mean B) were added to the Table 2 to improve data clarity.
- For eg., the authors mention that “Our results indicated that compounds 3b and 3c caused a significant decrease in cell viability in both melanoma lines and normal keratinocytes.” In fact, from table 2 compound 3a shows similar IC50 values.
- This manuscript fragment was corrected.
- As previously pointed out by the reviewer, the values compared by the authors (Table 4- 29.51% for 3b and 20.78 for IV) are comparable and not lower as the authors inferred.
- This manuscript fragment was corrected.
- Again, based on the results in Table 5, most values for compounds 3b, IV and V are also comparable. Unfortunately, the reviewer is not convinced that compound 3b shows significantly higher activity than its other analogs IV and V.
- This part of the manuscript has been revised. The reviewer's opinion has also been included in the added new fragment in the Conclusion. However, the authors did not suggest that compound 3b is significantly more active than its IV and V analogs, and used the term "slightly higher".
- Trifluoroacetic acid anhydride should be renamed as trifluoroacetic anhydride- line 86, 87.
- This was corrected.
Reviewer 2 Report
I consider that in this form the manuscript can be published.
Author Response
Dear Reviewer,
We are very grateful for the reviews of our manuscript.
This manuscript is a resubmission of an earlier submission. The following is a list of the peer review reports and author responses from that submission.
Round 1
Reviewer 1 Report
In this manuscript, the authors have synthesized a number of 5-trifluoromethyl-2-thioxo-thiazolo[4,5-d] pyrimidine derivatives and determined their anti-cancer activity in a number of cell lines. The authors have also determined the 3D structure of two of the analogs. Although the authors report a significant amount of work, it is not cohesive and well tied up towards their hypothesis. The hypothesis of this manuscript is not clear to the reader and the data are not well interpreted. Therefore, it is not recommended for publication.
Major:
- Firstly, the need to determine the crystal structures of the small molecules seems unnecessary for the purpose of the work. The compounds have been sufficiently characterized using H-, C-NMR, IR, MS, and melting point. There is no explanation on why the authors think it is important to know the orientation/3D conformation of these compounds.
Also, compound 3b was deemed the most promising based on its excellent antiproliferative activity, growth inhibition, etc. However, there is no crystal structure determined for compound 3b in the manuscript.
In short, there seems to be no rationale to incorporate this study.
- The authors mention that ‘As shown in Table 4 moderate growth inhibition of analog IV (-85.59%) and analog V (-88.95%) is lower than that of trifluoromethyl derivative 3b (-82.97%).’ These values are comparable and hence this statement is incorrect.
Again, based on the results in Table 5, most values for compounds 3b, IV, and V are also comparable. Unfortunately, the reviewer is not convinced that compound 3b shows significantly higher activity than its other analogs IV and V.
- Comparison of the newly synthesized analogs with the previous (non-trifluoro) analogs indicates that they showed similar activity to the previously reported ones. The authors also deduce that the Cl group might be responsible for the increased activity. However, there seems to be not much improvement if any due to the trifluoro-group. This has not been discussed.
- The compounds, 3b and others show cytotoxic effects in normal cell lines too. This has not been discussed in the manuscript.
Others:
- The synthetic route should be shown for step 1 in the synthesis.
- Trifluoroacetic acid anhydride should be renamed trifluoroacetic anhydride.
- The temperature of heating for synthetic route step 1 should be mentioned.
- Where is oxalyl chloride in the scheme? (Chlorination step)- do the authors mean phosphoryl chloride? It is not clear
- Key should be placed for all figures (Similar to figure 5)
- For better readability, please mention the type of cancers the respective cell lines belong to.
- Why was compound 4b not screened in the CCRF-CEM leukemia cell line?
- The following sentence should have Table 5 mentioned: “Analog V showed a good result, 58, 49 and 35 cell lines respectively, whereas IV demonstrated cytotoxicity (LC50) against only 3 cell lines.”
- IC50, LC50, etc. should be represented correctly (IC50, LC50)
- The manuscript would benefit from the editing of grammar and English writing.
Reviewer 2 Report
The manuscript pharmaceuticals-1470620 "Synthesis, Structural Characterization and Anticancer Activity of New 5-trifluoromethyl-2-thioxo-thiazolo[4,5-d]pyrimidine derivatives" by Lilianna Becan et al describe the synthesis, characterization, and biological activity of some derivatives of 3,4,5-trimetho5-trifluoromethyl-2-thioxo-thiazolo[4,5-d]pyrimidinexyphenyl. The synthesis of new compounds was confirmed by 1H, 13C NMR, IR spectroscopy, and elemental analysis. The cytotoxic and anticancer activities have also been evaluated.
The manuscript is well-written and presents a great interest to the readers of Pharmaceuticals.
As comments/suggestions:
1.How can one explain the superior antitumor activity of some compounds compared to others (ex compounds 3a-3d and 4a lines 183-201 or compounds presented in lines 232-239)?
2.What are the reference antitumors used to test the cytotoxic activity of compounds?
3. Why were only the 4 compounds (2b, 3b, 4b and 4c) selected by NCI?
4. What is the concentration of DMSO in the final solutions of the compounds used for the cell toxicity test?
line 359: "A mixture of appropriate thiazole 1 (a-e)" (quantity grams or moles?)
lines 396-399: reformulation
lines 435-437: "The selected amine (20 mM) was added to a suspension of 3b (10 mM)"(in what quantities or molar ratio?)